# eNAMPT Is Localised to Areas of Cartilage Damage in Patients with Hip Osteoarthritis and Promotes Cartilage Catabolism and Inflammation

**DOI:** 10.3390/ijms22136719

**Published:** 2021-06-23

**Authors:** Ashleigh M. Philp, Sam Butterworth, Edward T. Davis, Simon W. Jones

**Affiliations:** 1MRC-Versus Arthritis Centre for Musculoskeletal Ageing Research, Institute of Inflammation and Ageing, Queen Elizabeth Hospital, University of Birmingham, Birmingham B15 2WB, UK; as.philp@garvan.org.au; 2Division of Pharmacy and Optometry, School of Health Sciences, Manchester Academic Health Sciences Centre, University of Manchester, Manchester M13 9PL, UK; sam.butterworth@manchester.ac.uk; 3The Royal Orthopaedic Hospital NHS Foundation Trust, Bristol Road South, Northfield, Birmingham B31 2AP, UK; Edward.davis@nhs.net

**Keywords:** osteoarthritis, obesity, adipose tissue, adipokines, visfatin, NAMPT

## Abstract

Obesity increases the risk of hip osteoarthritis (OA). Recent studies have shown that adipokine extracellular nicotinamide phosphoribosyltransferase (eNAMPT or visfatin) induces the production of IL-6 and matrix metalloproteases (MMPs) in chondrocytes, suggesting it may promote articular cartilage degradation. However, neither the functional effects of extracellular visfatin on human articular cartilage tissue, nor its expression in the joint of hip OA patients of varying BMI, have been reported. Hip OA joint tissues were collected from patients undergoing joint replacement surgery. Cartilage explants were stimulated with recombinant human visfatin. Pro-inflammatory cytokines and MMPs were measured by ELISA and Luminex. Localisation of visfatin expression in cartilage tissue was determined by immunohistochemistry. Cartilage matrix degradation was determined by quantifying proteoglycan release. Expression of visfatin was elevated in the synovial tissue of hip OA patients who were obese, and was co-localised with MMP-13 in areas of cartilage damage. Visfatin promoted the degradation of hip OA cartilage proteoglycan and induced the production of pro-inflammatory cytokines (IL-6, MCP-1, CCL20, and CCL4) and MMPs. The elevated expression of visfatin in the obese hip OA joint, and its functional effects on hip cartilage tissue, suggests it plays a central role in the loss of cartilage integrity in obese patients with hip OA.

## 1. Introduction

Osteoarthritis (OA) is a leading cause of pain, disability, and shortening of adult-working life, and there are currently no approved disease-modifying OA drugs (DMOADs). In attempting to identify personalised DMOADs, it is now recognised that OA is a heterogeneous disease involving multiple tissues within the joint [1,2]. Importantly, obesity is a risk factor for OA, but the relationship between obesity and OA may not simply be due to weight-related increased loading on the joint, since BMI is also associated with the development of OA in non-weight-bearing joints, such as the hands [3,4]. One group of proteins that provide a link between obesity and joint damage are the adipose-secreted cytokines (adipokines), which are known regulators of metabolism [5,6,7,8,9,10] and the inflammatory response [11,12,13,14,15,16,17,18].

Previous research has demonstrated differential serum adipokine levels in relation to pathological indices associated with OA. In particular, Berry et al. (2012) found leptin levels to associate with bone formation markers, such as osteocalcin and PINP, and soluble leptin receptor with cartilage volume loss [19]; in turn, resistin was reported to mediate sclerotic bone formation in obese OA subchondral bone [15]. Furthermore, despite not finding adipokines to be associated with cartilage damage (IHC and proteoglycan content), de Boer et al. [11] (2012) suggest that adipokines may play important roles in OA development due to the heightened expression in OA patient serum and association with synovial tissue inflammation. Therefore, determining the expression and functional activity of adipokines could be a key biological effect area for identifying and developing a novel therapeutic target [11].

One adipokine that has received much attention in recent years is visfatin, a highly conserved, ubiquitously expressed protein, originally defined as a pre-B cell colony-enhancing factor [20], termed nicotinamide phosphoribosyltransferase (NAMPT or visfatin). Visfatin is an “adipokine-enzyme” due to its enzymatic nature when in a homodimeric conformation [21], and it is secreted independently from the Golgi apparatus and the endoplasmic reticulum from visceral adipose tissue [22]. Numerous potential roles for visfatin have now been proposed, including catalysing the conversion of nicotinamide and phosphoribosyl-pyrophosphates to nicotinamide mononucleotide [23], and acting as an insulin mimetic [24], a growth factor [25], or as an inflammatory cytokine able to induce TNF-α, IL-6, and IL-1β [26].

Visfatin exists in a compartmentalized fashion, with an intracellular version (iNAMPT) contributing to NAD+ biosynthesis through the conversion of nicotinamide into nicotinamide mononucleotide (NMN), allowing essential metabolic regulation [27]. Conversely, the extracellular version (eNAMPT) has received a lot of attention in current literature due to its elevated expression and potential contribution to a number of pathological conditions associated with ageing, including diabetes [28] and obesity [29,30,31,32]. Although the biosynthesis of nicotinamide adenine dinucleotide is well established as one of the functional roles of iNAMPT, the pathophysiological relevance and functional consequence of elevated levels of eNAMPT in disease states, including in OA, are not fully established.

Previous studies have shown that visfatin expression is increased in human chondrocytes upon stimulation with IL-1β and that the stimulation of chondrocytes with recombinant visfatin can modulate the expression of matrix proteases, aggrecan proteoglycan [33], and the production of IL-6 [21]. These studies suggest that eNAMPT may drive both pro-inflammatory and pro-degradative effects within joint cartilage. However, it is not known whether these effects of eNAMPT are mediated by modulation of NAD+ biosynthesis. Furthermore, these studies were conducted in isolated chondrocytes and not on articular cartilage tissue. This is important because isolated chondrocytes in culture rapidly de-differentiate, losing their Type II collagen, aggrecan proteoglycan phenotype, and upon proliferation adopt a more “fibroblast-like” morphology. In contrast, cartilage tissue chondrocytes have a rounded morphology, express high amounts of Type II collagen and aggrecan, and have no detectable proliferative activity [34,35,36]. In addition, although it is now well established that OA is a disease of the whole joint, there is little known about the expression of visfatin across the tissues of the OA joint, including the bone and synovium, nor how tissue expression varies with BMI in OA patients.

The aim of this study was therefore to examine the expression of visfatin locally within the tissues of the human hip OA joint, and to determine the functional and mechanistic role of visfatin in articular cartilage from patients with hip OA using a human ex-vivo cartilage explant model.

## 2. Results

### 2.1. Visfatin Is Expressed Locally by the Tissues of the Hip OA Joint and Is Elevated in the Synovial Tissues of Obese Patients

In reference to de Boer et al. [11] (2012), which suggested that adipokines may play important roles in OA development due to the heightened expression in OA patient serum, we first investigated the expression of eNAMPT in patient serum. In our OA patient cohort, there was no significant correlation between the systemic concentration of eNAMPT and BMI in patients with hip OA (Figure 1A).

Given that OA is predominantly a localised disease of the joint, we then investigated the localised expression of total visfatin in the tissues and synovial fluid of the hip OA joint. The presence of visfatin, under denaturing conditions, was examined in the synovial fluid as well as the tissues that encompass the hip joint, namely, cartilage, bone, skeletal muscle, synovium, and adipose, and compared to serum (Figure 1B,C). Electrophoretic mobility was confirmed with his-tagged recombinant visfatin. Notably, when compared w/w to cartilage, bone, and muscle, visfatin was found to be highly expressed in adipose tissue from the hip joint. Expression of visfatin was significantly greater in synovial fibroblasts (0.19 ± 0.05 vs. 1.65 ± 0.22, *p* = 0.02) from obese (OB) hip OA patients (*n* = 3), compared to normal weight (NW) hip OA patients (*n* = 3; Figure 1D,E). Visfatin expression in adipose tissue and synovial fluid displayed significant patient variability, but on average also appeared to be greater in OB patients compared to NW patients (Figure 1D,E).

### 2.2. Visfatin Induces the Production of Matrix Metalloproteases in Human Hip OA Cartilage and Is Co-Localised with MMP13 in Areas of Damage

Having observed that visfatin is expressed locally by the tissue of the OA joint, we next examined the effect of stimulating OA cartilage explants with recombinant visfatin (500 ng/mL for 24 h), compared to IL-1β, on the production of a panel of MMPs by Luminex. To this end, for each experimental condition, five cartilage explants were prepared from each of nine individual femoral head cartilage patient samples. Visfatin stimulation led to significant increases in a number of disease-relevant catabolic proteases, including MMP-1 (4-fold), MMP-2 (3-fold), MMP-3 (3-fold), MMP-7 (2.2-fold), MMP-8 (1.3-fold), MMP-9 (1.2-fold), MMP-10 (1.5-fold), and MMP-13 (5-fold) (Figure 2A–D). Visfatin’s role in cartilage degeneration was further investigated through IHC analysis of human hip OA femoral head sections. Staining for visfatin expression was more pronounced in areas of cartilage fibrillation and degeneration, when compared to areas of full-thickness cartilage located on the same femoral head, as shown in Figure 2E(i–iv). Furthermore, there was increased expression and co-localization of visfatin with MMP-13 in the pericellular extracellular matrix zone surrounding chondrocytes in areas of fibrillation, compared with chondrocytes in full-thickness cartilage (Figure 2F(i–vi)).

### 2.3. Visfatin Induces the Secretion of Pro-Inflammatory Cytokines and Chemokines in Human Hip OA Cartilage

We then examined the functional role of visfatin on hip OA cartilage cytokine production. This time, per condition, we prepared five cartilage explants from femoral head cartilage samples of *n* = 4 individual patients with hip OA and stimulated them with or without human recombinant visfatin (500 ng/mL) for 24 h. The effect of visfatin on the production of a panel of 44 known pro-inflammatory cytokines and chemokines was determined using Proseek technology. Following stimulation with visfatin, there was a notable increase in the production of 15 pro-inflammatory cytokines and chemokines, with significant increases in CCL4, MCP-1, and CCL20 in the tissue culture supernatant following visfatin stimulation compared to the media-only control (20-fold, 4-fold, and 7-fold, respectively) (Figure 3A). The effect of visfatin stimulation on the production of IL-6 was measured by ELISA in both NW (*n* = 4) and OB (*n* = 4) cartilage, compared to IL-1b stimulation. Stimulation of both NW and OB cartilage for 24 h with visfatin (500 ng/mL) led to a significant production in IL-6. However, the fold-change induction in the production of IL-6 was significantly much greater in OB cartilage explants compared to NW cartilage explants (7-fold vs. >1500-fold, respectively; *p* < 0.05) (Figure 3B).

### 2.4. eNAMPT Does Not Increase Intracellular NAD+ Production in Primary Human Hip OA Chondrocytes

Previous studies have attributed the functional effect of visfatin on IL-6 and MMP production in chondrocytes to modulation of intracellular NAMPT activity [21]. To investigate this, we stimulated primary hip OA chondrocytes with or without recombinant visfatin (i.e., eNAMPT) and quantified intracellular NAD+ production, as a measure of intracellular NAMPT activity. However, we observed no increase in NAD+ production following 24 h visfatin stimulation of primary chondrocytes (Figure 3C). Next, we examined whether the functional effect of recombinant visfatin on the induction of IL-6 secretion was dependent on intracellular NAMPT activity by co-stimulating the cells with a cell-permeable small molecule NAMPT inhibitor *N*-(4-((4-(phenylcarbamoyl)phenyl)sulfonyl)benzyl)imidazo[1,2-a]pyridine-6-carboxamide). However, despite the inhibitor significantly reducing NAD production (Figure 3D), we observed no blunting in the visfatin-mediated induction of IL-6, even at 1 µM concentration (Figure 3E). Therefore, the functional effect of visfatin in hip OA cartilage tissue appears unlikely to be mediated by changes in the enzymatic activity of intracellular NAMPT.

Given this finding, we then performed a visfatin receptor-binding identification screen using the Retrogenix cellular microarray platform (Retrogenix, Macclesfield, UK), which provided coverage of >2500 known human receptors. Following two confirmatory screens (Figure 3F,G), multiple hits were identified, including the bradykinin receptors BDKRB1 and BDKRB1, as well as CD44, GPRC5B, HCRTR2, LRP8, PEAR1, and CBL, suggesting that visfatin was capable of binding promiscuously to multiple receptors.

### 2.5. Visfatin Stimulates Loss of Proteoglycan and Exhibits Increased Expression and Co-Localisation with MMP13 in Areas of Cartilage Fibrillation

To determine whether the visfatin-mediated increase in the production of both pro-inflammatory cytokines and cartilage proteases was associated with cartilage proteoglycan degradation, we then measured the release of sulfated GAG (sGAG) from cartilage explants as a marker of proteoglycan loss. Stimulation of cartilage explants with visfatin (500 ng/mL) for 24 h induced an increase in the release of sGAG in both NW and OB cartilage explants, demonstrating that visfatin promoted proteoglycan loss (Figure 4). Of note, both the basal (non-stimulated) and visfatin-stimulated sGAG levels were higher in OB cartilage, compared to NW cartilage, suggesting that joint cartilage from obese patients with hip OA has a greater rate of proteoglycan degradation.

## 3. Discussion

This study is the first to report the functional effects of extracellular visfatin on human OA cartilage tissue, and to profile its expression and localisation in the joint tissues of patients with hip OA of varying BMI. Previously, it has been reported that visfatin is increased within the synovial fluid [37], infrapatellar fat pad, serum, and osteophytes [38] of patients with knee OA, compared to non-OA tissues. Importantly, we have now shown that visfatin is expressed locally by all the tissues of the hip OA joint (including cartilage, bone, synovium, and adipose) and that visfatin expression is elevated in the synovial fluid and synovial fibroblasts of obese hip OA patients compared to normal-weight hip OA patients. Furthermore, our histochemical analysis of OA femoral head sections reveals for the first time that visfatin expression is highly localized to areas of cartilage fibrillation, where it is co-localised with MMP13 in the pericellular extracellular matrix zone surrounding chondrocytes. Our finding that visfatin is elevated in the joint tissues of OA patients who are obese fits with previous studies that have reported that obesity imprints an inflammatory tissue phenotype. It is well known that obese adipose tissue is more inflammatory, with increased expression of pro-inflammatory cytokines and adipokines. However, we recently showed that obesity confers a more inflammatory phenotype on OA synovial tissue [39], with increased expression of IL-6 and IL-8. The mechanism by which obesity imprints this inflammatory phenotype is not known, but interestingly several long non-coding RNAs [40], which are known epigenetic regulators of gene expression, are differentially expressed in the obese state, and therefore may act to epigenetically confer an inflammatory phenotype. For example, we reported that the long non-coding RNA MALAT1 was upregulated in obese OA synovial tissue and regulated the production of IL-8 [39].

Our studies to examine the functional role of visfatin in human OA cartilage tissue provide the most significant findings of this paper and demonstrate that visfatin induces both pro-inflammatory and pro-degradative effects on hip OA cartilage explant tissue. Visfatin induced significant increases in the production of the collagenases MMP-1, MMP-8, and MMP-13, the gelatinases MMP-2 and MMP-9, stromelysins MMP-3 and MMP-10, and matrilysin MMP-7. Of all the collagenases, MMP-1 is most associated with newly formed collagen molecule breakdown, suggesting that visfatin may prevent cartilage repair as well as inducing cartilage resorption. Visfatin also induced a significant increase in the production of the gelatinase MMP-9. Gelatinases have remained largely under-researched in osteoarthritis. However, it was recently suggested that MMP-9 was fundamental to the activation of pro-MMP-13 [41], indicating that the partnership between MMP-9 and MMP-13 accelerated collagenase digestion. Therefore, our finding that visfatin induces the production of both MMP-9 and MMP-13 is notable.

Visfatin stimulation of hip OA cartilage explants also led to increased secretion of several pro-inflammatory mediators, suggesting that visfatin induces a profound inflammatory response in OA cartilage tissue. In particular, there was a significant increase in the production of IL-6, MCP-1, and CCL20, and also increased secretion of the chemokine ligand CCL4. Our finding that visfatin induces the production of CCL4 from articular cartilage is intriguing since it has previously been reported that CCL4 within OA synovial fluid is responsible for a large proportion of monocyte chemotactic activity [42]. This suggests that visfatin activity within the joint could play a key role in mediating the movement of monocytes into the synovial fluid, contributing towards synovitis. A key finding, however, was the differential response of normal-weight and obese OA articular cartilage tissue to visfatin stimulation, with significantly greater IL-6 production induced by visfatin in cartilage from obese hip OA patients. Pallu and colleagues (2010) previously noted a similar phenomenon in isolated cultured chondrocytes in response to leptin, where the leptin-mediated expression of TIMP2 and MMP-13 was dependent on the BMI of the patients from which the chondrocytes were isolated [43]. Furthermore, although in this study we did not measure the effect of visfatin on the activity of the aggrecanases (ADAMTS4/ADAMTS5), visfatin stimulation of cartilage led to a 3-fold increase in the release of sGAG, indicative of aggrecan proteoglycan loss and increased activity of the aggrecanases [44]. However, of significance, both basal and visfatin-stimulated production of sGAG was greater in articular cartilage from obese hip OA patients than in articular cartilage from normal-weight hip OA patients. These findings suggest that visfatin may mediate greater inflammatory-mediated cartilage damage in obese hip OA patients than in normal-weight hip OA patients. Since intracellular visfatin catalyses the production of NAD+, previous studies have attributed the functional effects of visfatin in chondrocytes to changes in NAD+ biosynthesis. However, in our study, we have demonstrated that stimulation of primary hip OA chondrocytes with extracellular visfatin does not affect intracellular NAD+ production, and thus the functional effects we have observed in hip OA cartilage appear unlikely to be due to changes in the enzymatic activity of intracellular visfatin. The receptor for visfatin has not been identified. However, our finding that visfatin is capable of binding to multiple human receptors, including CD44 and the bradykinin receptors BDKRB1 and BDKRB2, suggests that it likely acts promiscuously by interacting with multiple receptors, similar to many chemokine/cytokine ligands. Therefore, deconvoluting the intracellular signalling pathways that mediate the inflammatory and degradative effects of visfatin on cartilage likely represents a complex challenge.

It is important to note that all OA tissues in this study were received from patients undergoing joint replacement surgery and were therefore at an advanced stage of disease. The absence of joint tissue from early OA patients means we can only speculate about the potential role of visfatin in early OA disease initiation. Conducting functional studies on human joint cartilage from patients with early stage OA is inherently difficult due to the inaccessibility of sufficient tissue from patients not requiring joint surgery.

In conclusion, we have shown that the adipokine visfatin is expressed locally within the tissues of the hip OA joint and is co-expressed with MMP-13 in the pericellular zones of chondrocytes of fibrillated human OA cartilage tissue. Furthermore, we have shown that extracellular visfatin markedly induces both pro-inflammatory and pro-degradative effects on human hip OA cartilage tissue, particular in cartilage from obese patients. These data suggest that visfatin is a central mediator of cartilage degeneration in patients with hip OA. Targeted inhibition of visfatin signalling within the hip joint could therefore be a rewarding strategy for developing a novel therapeutic for obese patients with hip OA.

## 4. Materials and Methods

### 4.1. Participant Recruitment and Sample Collection

All experiments and methods were performed in accordance with the relevant guidelines and regulations. All experimental protocols were approved by a named institutional/licensing committee. Specifically, ethical approval was granted by the UK National Research Ethics committee (NRES 14-ES-1440), informed consent was obtained, and participants were recruited on a volunteer basis, after being fully informed of the study requirements by the clinical research staff of the collaborating hospitals. Patients with hip OA (age 45–80 years) undergoing elective total hip-joint replacement surgery (K and L grade 3–4) were recruited from The Royal Orthopaedic Hospital NHS Foundation Trust, Birmingham (UK), and Russells Hall Hospital, Dudley (UK). Serum was collected from *n* = 76 hip OA patients of varying adiposity and their BMI recorded. The femoral heads were collected and a portion of subcutaneous adipose tissue, gluteus maximus skeletal muscle, and synovium from around the joint collected following joint replacement surgery of *n* = 9 hip OA patients (patients were classified as either of normal weight (NW, BMI 18–24.9) (*n* = 4) or over-weight/obese (OB, BMI > 25) (*n* = 5)). Furthermore, synovial fluid was aspirated from *n* = 3 OB patients and from *n* = 3 NW patients. Patients who exhibited secondary causes of OA on the pre-operative radiograph were excluded from this study, for example developmental dysplasia, avascular necrosis, Perthes disease, slipped upper femoral epiphysis, and previous acetabular or femoral neck fractures. Patients receiving immunosuppressive therapy for inflammatory conditions or cancer, oral steroid treatment, and patients who have received an intra-articular steroid injection within 6 months were also excluded from this study.

### 4.2. Antibodies

The primary antibodies used in this study included anti-Visfatin (PA1-1045, Pierce Scientific, Hemel Hempstead, UK), α-NFκB (SC-8008, Santa Cruz Biotech, Santa Cruz, CA, USA), and anti-MMP-13 (11365013, Thermo Scientific, Hemel Hempstead, UK). The secondary antibodies used in this study were Alexa Fluor^®^ 555 anti-rabbit H + L (A-21428), Alexa Fluor^®^ 488 anti-mouse IgG1 (A-21121), and DAPI (D3571) (Life Technologies, Warrington, UK).

### 4.3. Serum Visfatin Profiling by ELISA

Serum visfatin expression was determined using a Visfatin EIA kit (RAB0377; Sigma Aldrich, St. Louis, MO, USA).

### 4.4. Preparation of Cartilage Explants and Isolation of Primary Synovial Fibroblasts

Cartilage was separated from the bone of the patients’ femoral head. Explants discs (3 mm) were prepared using a cork-borer and placed into a 96-well tissue culture plate containing chondrocyte growth media (10% FBS, 100 U/mL penicillin streptomycin, 2 mM L-glutamate, 1% non-essential amino acids, and 2.5 mg/mL amphotericin B). For the isolation of primary synovial fibroblasts, synovium tissue was digested with 2 mg/mL Collagenase Type 1A (Sigma Aldrich, USA) for 5 h at 37 °C. The sample was then filtered, and cells resuspended in growth media. The remaining synovium and cartilage, together with bone, adipose, and muscle joint tissues, were snap-frozen in liquid nitrogen and powdered (Spex Sample Prep 6770 freezer mill; Stanmore, UK) for protein analysis. The samples’ protein content was determined by BCA assay (Thermo Fisher, Hemel Hempstead, UK) and equal amounts were loaded onto a Western blot. Equal loading was confirmed by ponceau.

### 4.5. Explant Stimulation

Prior to stimulation, explants were cultured for 7 days to prevent bias from cutting. Explants were stimulated with 500 ng/mL human recombinant visfatin sourced from *E. coli* (Biovision, Milpitas, CA, USA) or 1 ng/mL recombinant human IL1β sourced from *E. coli* (Sigma Aldrich, USA) in 1% serum chondrocyte media (1% FBS, 100 U/mL penicillin streptomycin, 2 mM L-glutamate, 1% non-essential amino acids, and 2.5 mg/mL amphotericin B) for 24 h, unless otherwise stated.

### 4.6. Determination of Matrix Metalloproteinase (MMP) and Proinflammatory Cytokine Production

IL-6 production was assessed in explants using an IL-6 Duoset ELISA (DY206, RnD systems, Abingdon, UK). MMP secretion from human cartilage explants was determined by using an MMP 9-plex Luminex assay kit (Biorad, Hercules, CA, USA). Cytokine and chemokine production was assessed using the Proseek Multiplex Inflammation kit (Olink, Uppsala, Sweden).

### 4.7. Synthesis of the NAMPT Inhibitor N-(4-((4-(Phenylcarbamoyl)phenyl)sulfonyl)benzyl)imidazo[1,2-a]pyridine-6-carboxamide

The compound was prepared using a previously described synthetic sequence and conditions [45].

### 4.8. Determination of Sulphated Glycoaminoglycan (sGAG)

Sulphated glycosaminoglycan (sGAG) from cartilage explant supernatants was measured as an indicator of aggrecan degradation via a dimethylmethylene blue (DMMB) assay [46]. Shark chondroitin C was used as a standard control.

### 4.9. Immunohistochemistry of the Femoral Head

Femoral heads were decalcified in 5% formic acid at room temperature and embedded into paraffin. Slides were dewaxed and rehydrated in a xylene and ethanol series and washed in PBS. Samples were heated as free-floating sections in 10 mM sodium citrate (pH 8.5, 60 min, 80 °C) with gentle agitation prior to staining. Free floating sections were blocked in 10% v/v goat serum in phosphate buffer and 0.3% Triton-X 100 and incubated overnight at 4 °C in primary antibodies. All primary antibodies were used at their optimal concentrations, which were determined empirically. Detection was achieved using Alexa-conjugated secondary antibodies, and sections mounted using Prolong R Gold Antifade mountant (Life Technologies, Warrington, UK). Images were obtained using a Zeiss Axiovert UV confocal microscope and Zeiss Zen 2010 software. H&E staining was performed on de-paraffinized and rehydrated sections. Sections were stained in Mayer hematoxylin (Sigma Aldrich, Gillingham, UK) for 8 min, and counterstained in eosin Y (Sigma Aldrich, Gillingham, UK).

### 4.10. NAD Activity Assay

Primary human hip chondrocytes (6 × 103 cells per well) were plated in an opaque white-walled 96-well tissue culture plate (Corning R 3917) and treated with NAMPT small molecule inhibitor for 1 h (1 μM, 10 nM, or 0.1 nM) prior to co-incubation with recombinant visfatin (500 ng/mL) for 24 h in a humidified atmosphere of 37 °C and 5% CO_2_. The NAD/NADH-Glo assay (Promega Corporation, Madison, WI, USA) detects oxidised and reduced nicotinamide adenine dinucleotides. The assay was used as per the manufacturer’s instructions. Briefly, 25 μL of NAD/NADH-Glo detection reagent (Luciferin Detection Reagent, Reductase, Reductase Substrate, NAD cycling Enzyme, NAD Cycling Substrate) was added to each well containing 25 μL of chondrocyte growth media. The plate was agitated for 5 min and incubated at RT for a further 60 min. Luminescence was recorded at 30 and 60 min using a luminometer (Centro LB 960, Berthold Technologies, Bad Wildbad, Germany).

### 4.11. Receptor Identification Screen

In order to identify the candidate receptors for visfatin, a binding screen of his-tagged recombinant visfatin against >2500 human membrane proteins, was performed using the Retrogenix Cell Microarray platform (Macclesfield, UK). In brief, binding conditions were optimised for binding of his-tagged recombinant proteins. Expression vectors encoding each of the human membrane proteins were spotted onto glass slides. A HEK293 cell monolayer was cultured over the glass slide, resulting in overexpression of each of the human membrane proteins via reverse transfection. In the primary screen, slides were incubated with 2.5 ug/mL his-tagged visfatin or his-tagged EGF (control), and receptor interactions were detected using a mouse anti-His antibody (Millipore) followed by an AlexaFluor^647^ anti-mouse antibody (Life Technologies). Receptor hits were identified by visual inspection using ImageQuant software (GE Healthcare, Chicago, IL, USA). Following the primary screen, vectors encoding each of the positive hits were sequenced and 2 confirmatory screens performed. The first confirmatory screen used his-tagged ligands followed by AlexaFluor^647^ anti-His detection. The second confirmatory screen was performed using his-tagged ligands attached to AlexaFluor^647^ and anti-His-coated beads (high sensitivity) were used to confirm specificity.

### 4.12. Statistical Analysis

Descriptive statistics were tabulated to detail the patients’ characteristics (mean ± SD). Mann–Whitney U tests and Wilcoxon Signed-Rank tests (Graphpad Prism) were used to make comparisons. A *p* < 0.05 was accepted as statistically significant. All graphs are shown as the mean ± SEM unless otherwise stated.

## Figures and Tables

**Figure 1 ijms-22-06719-f001:**
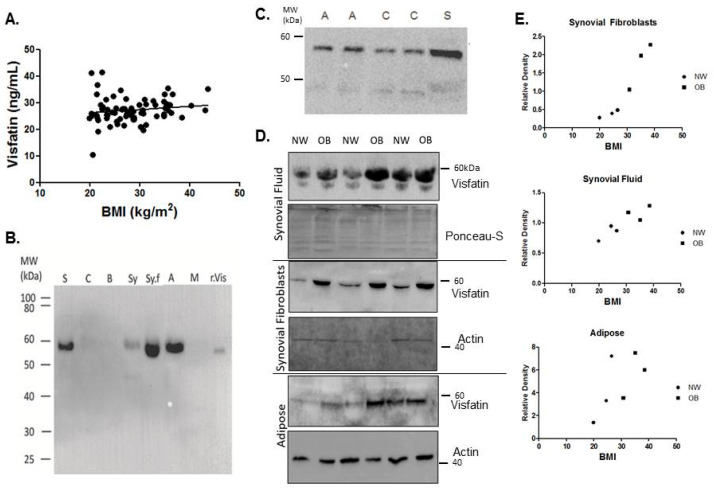
The expression of visfatin in the serum and joint tissues of patients with hip OA. (**A**) Serum extracellular visfatin concentrations were determined by ELISA in *n* = 76 hip OA patients of varying BMI. (**B**) Expression of total visfatin under reducing conditions in w/w matched tissue samples (serum (S), cartilage (c), bone (B), synovium (Sy), synovial fluid (Sy.f), infrapatellar fat pad (IFP), adipose (A), and muscle (M)) (*n* = 3 in total, representative blots shown). Recombinant visfatin (r.Vis) indicates the molecular weight region. (**C**) Confirmation of detection of visfatin in cartilage. Cartilage visfatin expression was confirmed through increasing cartilage (C) protein load and reducing adipose (A) and serum (S) protein load. (**D**) Tissue panel of a Western blot sample from normal-weight (NW) and obese (OB) individuals with hip OA. All samples were normalised to μg of total protein loaded, and an equal loading was confirmed by ponceau-S staining and actin expression. (**E**) Western blots were analysed using Image J software and the densitometry was compared in NW and OB patients. Relative density was calculated according to the protein loading control and plotted against patient BMI.

**Figure 2 ijms-22-06719-f002:**
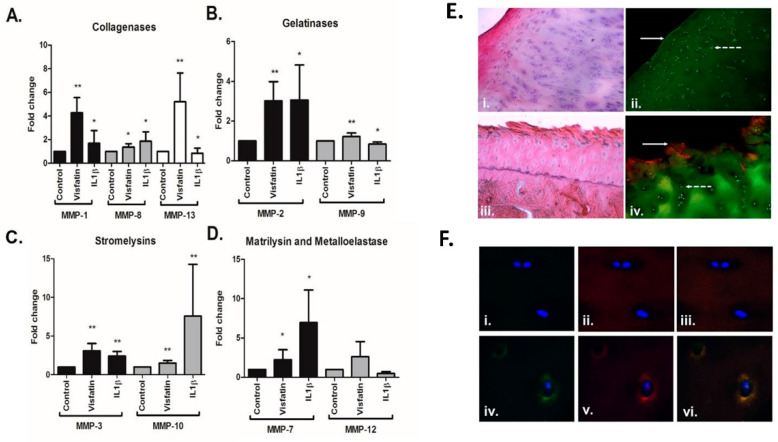
Visfatin induces the production of cartilage catabolic proteases in hip OA cartilage and is co-localised with MMP13 at sites of damage. MMP secretion from cartilage explants following visfatin stimulation (500 ng/mL) and IL-1ß (1 ng/μL) using Luminex, and separated into the MMP classification (*n* = 9 individual patients (five explants per patient)). (**A**) Collagenases classification. (**B**) Gelatinases classification. (**C**) Stromelysins classification. (**D**) Matrilysins and Metalloelastase classification. * = *p* < 0.05, ** = *p* < 0.01, significant difference between the treatment and control values. (**E**) IHC of cartilage on a human femoral head showing full-thickness and fibrillated cartilage: (i) H&E staining of full-thickness cartilage; (ii) fluorescent images of full-thickness cartilage (anti-visfatin shown in red, anti-NFkB shown in green; *n* = 4 individual patients)—solid arrow represents area of smooth cartilage, and the dotted arrow demonstrates low visfatin expression within the pericellular area of the chondrocytes; (iii) H&E staining of degraded and fibrillated cartilage (25× magnification); (iv) fluorescent images of degraded and fibrillated cartilage (anti-visfatin shown in red, anti-NFkB shown in green; *n* = 4 individual patients) (25× magnification)—solid arrow represents area of smooth cartilage, and the dotted arrow demonstrates increased visfatin expression within the pericellular area of chondrocytes. (**F**) Co-expression of MMP-13 and visfatin in degraded and fibrillated cartilage (63× magnification): (i) MMP-13 expression in chondrocytes of full-thickness cartilage; (ii) visfatin expression in chondrocytes of full-thickness cartilage; (iii) co-staining of visfatin and MMP-13 in chondrocytes of full-thickness cartilage; (iv) MMP-13 expression in chondrocytes of degraded cartilage; (v) visfatin expression in chondrocytes of degraded cartilage; (vi) co-staining of visfatin and MMP-13 in chondrocytes of degraded cartilage.

**Figure 3 ijms-22-06719-f003:**
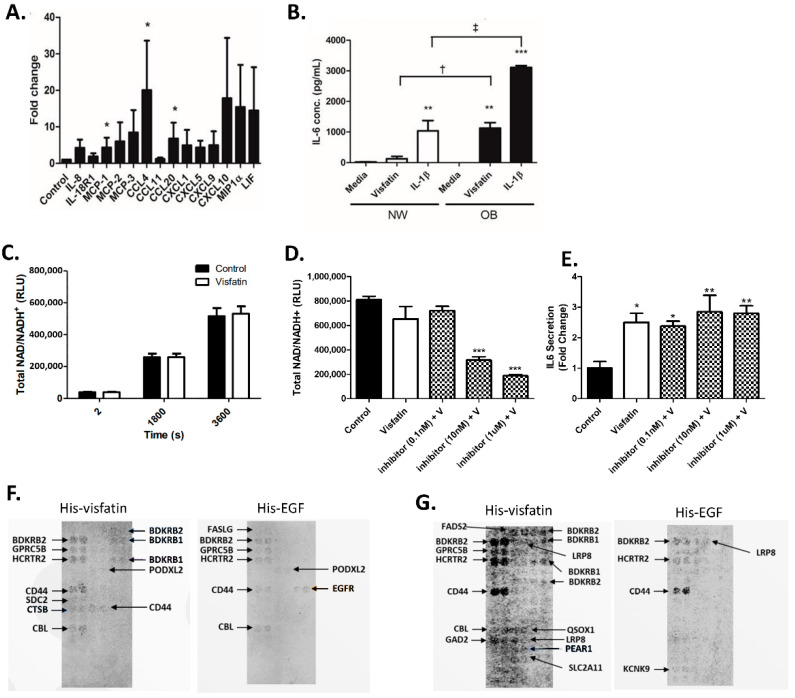
Visfatin induces the production of pro-inflammatory cytokines in hip OA chondrocytes independent of NAD+ production. (**A**) 24 h cytokine production in tissue culture supernatants following visfatin-stimulated cartilage explants. Cytokines and chemokines were measured using Proseek technology in tissue culture supernatants of the media and visfatin-stimulated (500 ng/mL) explants (*n* = 4 biological replicates). (**B**) Production of IL-6 following visfatin and IL-1ß stimulation of NW (*n* = 4) and OB (*n* = 4) cartilage. Five explants were stimulated from *n* = 4 individual patients per BMI group. * = *p* < 0.05, ** = *p* < 0.01, *** = *p* < 0.001—significant difference between the treatment and control values. † = *p* < 0.01, ‡ = *p* < 0.001—significant difference between the NW and OB values. (**C**) Primary OA chondrocyte NAD+ production over 2–3600 s following cell lysis following the addition of recombinant visfatin. (**D**) Effect of the NAMPT inhibitor *N*-(4-((4-(phenylcarbamoyl)phenyl)sulfonyl)benzyl)imidazo[1,2-a]pyridine-6-carboxamide at 0.1 nM, 10 nM or 1 µM on NAD+ production and (**E**) on recombinant visfatin-mediated (500 ng/mL) induction of IL-6 secretion in primary OA chondrocytes. * = *p* < 0.05, ** = *p* < 0.01, *** = *p* < 0.001—significant difference between the treatment and control values. (**F**) Confirmatory receptor identification screen using his-tagged visfatin and his-tagged EGF (control), followed by AlexaFluor^647^ anti-His detection. (**G**) High sensitivity confirmatory receptor identification screen using his-tagged ligands attached to AlexaFluor^647^ anti-His-coated beads.

**Figure 4 ijms-22-06719-f004:**
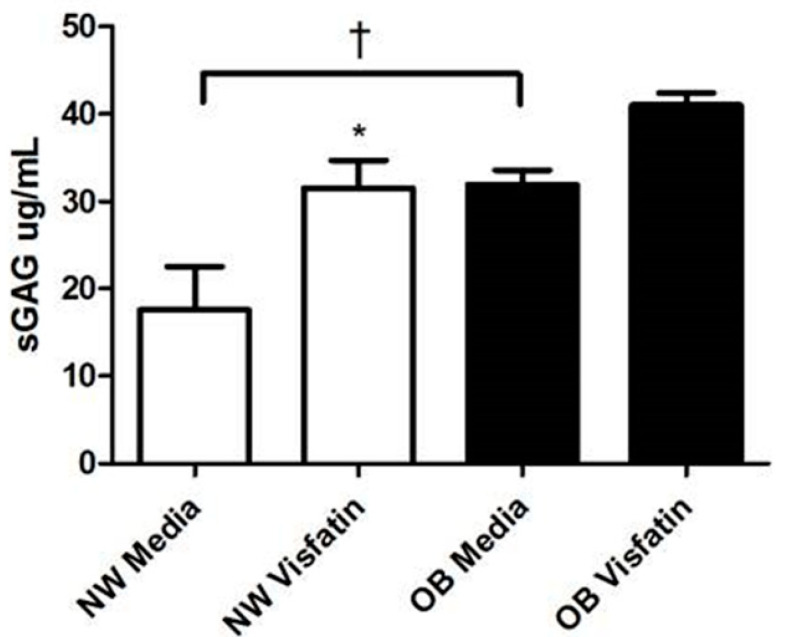
Visfatin induces cartilage proteoglycan degradation. Sulphated glycosaminoglycan (sGAG) secretion into tissue culture supernatants following visfatin stimulation of human cartilage explants from normal weight (NW) and obese patients (OB) (*n* = 4 individual patients, with five explants stimulated per patient) * = *p* < 0.05, significant difference between the treatment and control values. † = *p* < 0.05, significant difference between the NW and OB values.

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
