# Peer review of "eNAMPT Is Localised to Areas of Cartilage Damage in Patients with Hip Osteoarthritis and Promotes Cartilage Catabolism and Inflammation"

_ijms, 2021, doi:10.3390/ijms22136719_

Round 1

Reviewer 1 Report

The paper is very interesting and properly elaborated, but before acceptation for publication it needs some minor revisions that are presented below:

  1. In title of the article instead of abbreviation eNAMPT a full name of this compound should be applied
  2. The conclusions should be presented in form of a separate chapter
  3. The references should be presented in uniform style as in actual version of the manuscript some titles of Journal are presented using abbreviations and the others with the use of full names.

Author Response

The paper is very interesting and properly elaborated, but before acceptation for publication it needs some minor revisions that are presented below:

Thank you for the positive review of our manuscript and the insightful comments.

1. In title of the article instead of abbreviation eNAMPT a full name of this compound should be applied

Thank you for this suggestion. We have now added the full name for eNAMPT into both the abstract (line 14) and introduction (line 54). However, inclusion of the full abbreviation “extracellular nicotinamide phosphoribosyltransferase” or eNAMPT would make our title too long to meet the journal requirements.

2. The references should be presented in uniform style as in actual version of the manuscript some titles of Journal are presented using abbreviations and the others with the use of full names.

Apologies for this error. This has now been amended in the revised manuscript.

Reviewer 2 Report

Authors attempted to examine that up-regulation of visfatin expression could promote cartilage degeneration, resulting in the development of obese hip OA joint. Although their approaches are interesting and have some impacts to people who deal with OA patients, the manuscript lacks direct evidence of mechanisms how visfatin can induce cartilage degeneration and how the obesity can relate to enhanced expression of visfatin in OA tissues.

Major concerns:

1) Authors presented relationship between visfatin expression and the development of OA in this manuscript. However, direct effect of visfatin on cartilage degeneration was obscure. Authors must examine inhibition experiments, using visfatin inhibitors or knockdown of visfatin gene by siRNAs to emphasize that upregulated expression of visfatin can induce tissue damage . If results from the inhibition tests of visfatin showed suppression of cartilage degeneration, the readers understand the direct effect of visfatin on the development of OA.

2) Please more explain why and/or how the obesity can enhance visfatin expression in the discussion section.

Minor concern

1) In the legend of Figure 1C, authors must add to describe the meaning of “A”, “C”, and “S”. 

Author Response

1. Authors presented relationship between visfatin expression and the development of OA in this manuscript. However, direct effect of visfatin on cartilage degeneration was obscure. Authors must examine inhibition experiments, using visfatin inhibitors or knockdown of visfatin gene by siRNAs to emphasize that upregulated expression of visfatin can induce tissue damage . If results from the inhibition tests of visfatin showed suppression of cartilage degeneration, the readers understand the direct effect of visfatin on the development of OA.

We have already utilised a small molecule inhibitor of visfatin, namely (N-(4-((4-(phenylcarbamoyl)phenyl)sulfonyl)benzyl)imidazo[1,2-a]pyridine-6-carboxamide) in the study for the experiments detailed in Figure 3D and 3E.

This small molecule inhibitor acts to inhibit the intracellular enzymatic role of visfatin, which converts nicotinamide into nicotinamide mono-nucleotide (NMN) thus supporting NAD+ biosynthesis. The purpose of these studies therefore was to examine whether this intracellular enzymatic role of visfatin was responsible for the functional effect of visfatin on chondrocytes. Our data showed that despite inhibition of intracellular activity we still observed that stimulation of chondrocytes with visfatin induced IL-6 expression. Therefore, our data shows that the functional effect of extracellular visfatin in promoting cartilage inflammation is not dependent on the enzymatic intracellular role of visfatin. Instead, our data suggests that the effects of visfatin on chondrocytes/cartilage as via an as yet unidentified receptor. 

To address this potential mechanism we have now included additional experimental data in the revised manuscript (Figure 3F and 3G), where we have conducted a receptor binding screen of > 2500 human membrane proteins (Retrogenix Microarray platform) to identify candidate visfatin receptors using a his-tagged visfatin compared to a his-tagged EGF protein (control). Receptor interactions were detected using an AlexaFluor 647 antibody and visual inspection using ImageQuant software. Both a primary and a confirmatory screen was performed and a number of hits were identified. These data demonstrate that visfatin is a “promiscuous” ligand, in that it can bind to a multiple cell surface receptors in a similar way to many chemokine and cytokine ligands. Therefore, deconvoluting the downstream signalling pathways via these multiple receptors is highly challenging and unlikely to lead to a specific targeted approach for therapeutics.

In the revised manuscript we have included new data in Figure 3F and 3G, and discuss this point in the revised discussion. Lines 329-335.

“The receptor for visfatin has previously not been identified. However, our finding that visfatin is capable to binding to multiple human receptors, including CD44 and the bradykinin receptors BDKRB1 and BDKRB2, suggests that it likely acts promiscuously by interacting with multiple receptors similarly to many chemokine/cytokine ligands. Therefore, deconvoluting the intracellular signalling pathways that mediate the in-flammatory and degradative effects of visfatin on cartilage likely represents a complex challenge”. 

2. Please more explain why and/or how the obesity can enhance visfatin expression in the discussion section.

The reviewer raises a good point, which we have now addressed in the discussion of the manuscript, lines 276-287

“Our finding that visfatin is elevated in the joint tissues of OA patients who are obese fits with previous studies that have reported that obesity imprints an inflammatory tissue. Phenotype. It is well known that obese adipose tissue adopts a more inflammatory phenotype, with increased expression of pro-inflammatory cytokines and adipokines. However, we recently showed that obesity confers a more inflammatory phenotype on OA synovial tissue [40], with increased expression of IL-6 and IL-8. The mechanism by which obesity imprints this inflammatory phenotype is not known, but interestingly several long non coding RNAs [41], which are known epigenetic regulators of gene ex-pression, are differentially expressed in the obese state, and therefore may act to epi-genetically confer an inflammatory phenotype. For example, we reported that the long non-coding RNA MALAT1 was upregulated in obese OA synovial tissue and regulated the production of IL-8 [40].”

3. In the legend of Figure 1C, authors must add to describe the meaning of “A”, “C”, and “S”. 

The meaning of A, C and S has been added to the Figure 1C legend in the revised manuscript.

Round 2

Reviewer 2 Report

The issues addressed have been resolved in this revised manuscript.

Author Response

There were no further comments to address from the reviewer